# An Overview of the Antioxidant Effects of Ascorbic Acid and Alpha Lipoic Acid (in Liposomal Forms) as Adjuvant in Cancer Treatment

**DOI:** 10.3390/antiox9050359

**Published:** 2020-04-25

**Authors:** Mohamed Attia, Ebtessam Ahmed Essa, Randa Mohammed Zaki, Amal Ali Elkordy

**Affiliations:** 1School of Pharmacy and Pharmaceutical Sciences, Faculty of Health Sciences and Wellbeing, University of Sunderland, Sunderland SR1 3SD, UK; bh63qn@research.sunderland.ac.uk; 2Department of Pharmaceutical Technology, Pharmacy College, Tanta University, Tanta 31111, Egypt; ebtesam.eisa@pharm.tanta.edu.eg; 3Department of Pharmaceutics, Faculty of Pharmacy, Prince Sattam Bin Abdulaziz University, Alkharj 11942, Saudi Arabia; randazaki439@yahoo.com; 4Department of Pharmaceutics and Industrial Pharmacy, Faculty of Pharmacy, Beni-Suef University, Beni-Suef 62511, Egypt

**Keywords:** liposomes, ascorbyl palmitate, reactive oxygen species, doxorubicin, alpha-lipoic acid, ascorbic acid

## Abstract

Antioxidants are known to minimize oxidative stress by interacting with free radicals produced as a result of cell aerobic reactions. Oxidative stress has long been linked to many diseases, especially tumours. Therefore, antioxidants play a crucial role in the prevention or management of free radical-related diseases. However, most of these antioxidants have anticancer effects only if taken in large doses. Others show inadequate bioavailability due to their instability in the blood or having a hydrophilic nature that limits their permeation through the cell membrane. Therefore, entrapping antioxidants in liposomes may overcome these drawbacks as liposomes have the capability to accommodate both hydrophilic and hydrophobic compounds with a considerable stability. Additionally, liposomes have the capability to accumulate at the cancer tissue passively, due to their small sizes, with enhanced drug delivery. Additionally, liposomes can be engineered with targeting moieties to increase the delivery of chemotherapeutic agents to specific tumour cells with decreased accumulation in healthy tissues. Therefore, combined use of liposomes and antioxidants, with or without chemotherapeutic agents, is an attractive strategy to combat varies tumours. This mini review focuses on the liposomal delivery of selected antioxidants, namely ascorbic acid (AA) and alpha-lipoic acid (ALA). The contribution of these nanocarriers in enhancing the antioxidant effect of AA and ALA and consequently their anticancer potentials will be demonstrated.

## 1. Introduction

Reactive oxygen species (ROS) are normal products of the cell aerobic metabolic reaction. They contain oxygen in the form of peroxides and superoxide hydroxyl radicals, singlet oxygen or hydrogen peroxide. ROS can be produced at elevated amounts under pathophysiological conditions. ROS is generally induced endogenously via ROS-generating enzymes, such as xanthine oxidase and metabolic by-products formed by the electron transport chain reaction. Externally, many factors including environmental stress, such as exposure to ionising radiation or excess ultra violet (UV) radiation, can increase ROS production. ROS may cause damage to cell membranes, lipids, proteins and DNA, causing serious damage and impairment in their normal functions. This may lead to mutations, apoptosis and failure within these systems [1]. Oxidative stress, the imbalance between the production of ROS and antioxidant protection mechanisms, therefore originates from the inability of endogenous antioxidant defence mechanisms to protect against these impairments. This would result in the development and aggravation of many disease conditions such as diabetes [2], Parkinson’s disease [3], Alzheimer’s disease [4], acute renal failure [5], lung failure [6] and cancer [7]. Therefore, administration of antioxidant supplements is recommended to reduce oxidative damage to the human body. Antioxidants generally exert their effects mainly by either preventing the production of ROS or scavenging the formed ROS. Certain types of antioxidants exert their activity by degrading ROS into less harmful or neutral products [8]. In cancer treatment, chemotherapy induces an increase in reactive oxygen species (ROS) production in cancerous cells [4]. Antioxidants exert a major effect in treating and protecting against cancer.

ROS effects can be double sided, where they can kill both normal and cancerous cells by damaging proteins, lipids and DNA or even induce cancer [9,10,11]. In contrast, ROS manipulation can induce apoptosis to the cancer cell only because normal cells have a different redox environment compared to cancer cells and are less sensitive to redox manipulation [12]. Therefore, ROS modulation using antioxidants or pro-oxidants is a promising strategy to selectively target cancer cells during chemotherapy treatment [13,14,15].

Many chemical compounds have been examined for their antioxidant properties. Based on their origin, these antioxidants may be either endogenous (e.g., glutathione and uric acid) or exogenous. Generally, the majority of antioxidants come from our diet [16]. Natural antioxidants have the ability to potentially modulate oxidative stress.

Over the last decades, researchers have been focusing on producing some promising cytotoxic and anticancer drugs originating from natural compounds such as alpha lipoic acid, ascorbic acid, curcumin and many other compounds. They were mainly focused on developing some novel therapeutic strategies as alternative drugs to conventional chemotherapy, to reduce or eliminate the side effects of the current chemotherapy or to potentiate a synergetic effect with chemotherapeutics. This initiative was mainly taken to overcome the major side effects of conventional therapy, improve the patient compliance and reduce the cost [17,18,19].

There are many comprehensive reviews about nano-antioxidants and their classification [8,20]. Additionally, many research articles are available on the delivery of antioxidants in nanocarrier systems for enhancing the efficiency of antioxidant agents in the presence or absence of cytotoxic agents. This review will focus mainly on the liposomal delivery of two of the most widely investigated antioxidants, ascorbic acid an alpha lipoic acid, for which there is sparse publication.

## 2. Role of Antioxidants in Cancer Therapy

Once free radicals are formed, they are capable of disrupting the cell metabolic pathway and structure, leading to formation of more free radicals. This can, in turn, lead to more extensive cell and tissue damage. Uncontrolled free radical production is considered as one of the major factors in many diseases. Antioxidant therapy can be defined as treatment that prevents or reduces the side effects of free radicals. One of the strategies to modify oxidant-mediated cell injury is based on increasing the antioxidant capacity of cells or inhibiting the production of ROS. Antioxidants exert a chemo-preventive effect and improve chemotherapy effectiveness. The effectiveness of exogenous antioxidants to protect tissues from oxidative stress in vivo is variable and depends on the type of the antioxidant, its biopharmaceutical properties, its concentration at the site of action and the nature of the oxidative stress [21,22]. The following section will present brief information about ascorbic acid (AA) and alpha-lipoic acid (ALA).

## 3. Ascorbic Acid (AA)

Ascorbic acid “AA” (Figure 1a) is a water-soluble antioxidant and enzyme co-factor produced by plants and certain animals. AA is an essential micronutrient antioxidant with a powerful reducing effect that plays an essential role in numerous physiological processes in the human body. AA helps in the synthesis and metabolism of other vitamins and amino acids required by the body such as tyrosine, folic acid, lysine and tryptophan. It also helps in increasing the absorption of iron by the body by reducing ferric to the ferrous state [23,24,25]. 

It has been reported that AA is useful in prevention and treatment of cancer based on both in vivo and in vitro studies. AA causes apoptosis and inhibition of cell proliferation by its ability to regulate the cellular redox state [18,24,26]. In the 1970s, Cameron and co-workers were the first to report the proposed anticancer activity of AA, only at high doses. They conducted a series of studies on the anti-tumour effects of AA, and concluded that AA extended the lifespan of cancer patients with enhanced quality of life [27,28]. Nevertheless, other investigators failed to obtain similar results even when using very high doses (up to 10 g daily) administered orally in pills. Other investigators even suggested that an overdose of vitamins may even hold a risk of cancer initiation [29,30]. Such contradicting findings opened the subject of the therapeutic effectiveness of AA to a wide debate. It was not until the early 1990s that AA started to be accepted by the medical profession as having a potential role in cancer prevention [31]. Shortly after that, it was documented that AA had anticancer effect only if it was taken by intravenous (40–400 mg/kg) or intraperitoneal (4 g/kg) injections, where it can reach the blood at high concentration compared to the oral route [32,33,34]. Since then, several in vitro studies were conducted to evaluate the role of AA in the prevention and/or treating different types of cancer cell lines. However, in vivo trials are limited because of the high blood level required to obtain the intended effect.

The mechanism of action of AA is still hotly debated and under discussion, where it acts by different and contradictory mechanisms of action depending on the dose and frequency of administration [35].

A low concentration of AA was observed to act as an antioxidant. Previous research reported that low concentrations of AA reduced ROS and removed oxidative stress. This helped in maintaining the intracellular redox balance and minimised the free radicals that caused cell damage. Additionally, it was also documented that AA could neutralise the ROS produced from the imbalance between intrinsic antioxidant and oxidative stress. Additionally, it has a role in modulating Ki67 expression, decreasing inflammation through decreasing C-reactive protein and pro-inflammatory cytokines. Therefore, AA plays a potential role in decreasing tumour growth in pancreas, breast, kidney, lung and liver cancers [24,36,37,38,39].

At a higher AA concentration, AA is more toxic towards cancer cells because of the increased uptake of its oxidised form, dehydroascorbate, via the glucose transporters (GLUT1 and GLUT3) [40,41]. Due to transferrin receptor overexpression in cancer tissue, a high number of cytoplasmic iron ions from ferritin, which become available for dehydroascorbate, result in production of a significant amount of H_2_O_2_ that preferentially targets cancer cells. This is in addition to the depletion of reduced glutathione by the oxidation–reduction mechanism, leading to cellular damage, cell death and apoptosis [35,37,42].

## 4. Alpha-Lipoic Acid (ALA)

Alpha-lipoic acid (ALA), also known as thioctic acid, is a naturally occurring, short-chain fatty acid, which contains sulphur in its structure (Figure 1b). It has been used in managing many disease conditions such as diabetes mellitus, hypertension, Alzheimer’s disease, Down syndrome, cognitive dysfunction and some type of cancers including breast cancer [43]. The use of ALA as an alternative medicine has been growing fast as a therapeutic agent and nutritional supplement [19,44]. ALA’s antioxidant effect plays a major role in cellular growth due to its ability to scavenge ROS and renew endogenous antioxidants. ALA is reduced to dihydrolipoic acid (DHLA) that shows a unique characteristic as free radical scavenger and modifies many oxidative stress and inflammatory pathways [44]. Previous studies showed that the intercellular redox balance is linked with the cellular growth and plays a critical role in carcinogenesis. ALA was proposed to reduce elevated oxidative stress accumulated by cancerous cells, leading to apoptosis and inhibition of cell proliferation [45,46,47,48].

## 5. Roles of Ascorbic Acid (AA) and Alpha Lipoic Acid (ALA) in Cancer Therapy

Many researchers have stated that antioxidants such as AA and ALA have substantial roles in potentiating and synergising the cytotoxic effect of antineoplastic agents while protecting normal body tissue. This reduces chemotherapy side effects and increases patient survivability [9,13].

Anti-tumour effects of both AA and ALA were noticed by cell cycle arrest at the G1 phase through the increment of p53 protein [17,37,49,50]. p53 is a tumour suppressor factor that has a significant role in modulating cell-cycle checkpoints, DNA repair and apoptosis. Additionally, p53 has been associated with ROS generation and ROS-induced oxidative stress [11,51]. Both antioxidants have been shown to selectively increase p53 gene expression in tumour cells compared to the neighbouring normal cells, since tumour cells have been known to be down-regulated and deficient from p53 [52,53].

Immunohistochemical assessment of a proliferative activity using Ki-67 antigen expression revealed that both AA and ALA treatments showed significantly decreased Ki-67 expression in breast cancer cells compared to the untreated control [37,54,55,56,57]. Ki-67 is considered to be a tumour proliferation marker present in different phases of the cell cycle in the cell nucleus, and it has been used as prognostic factor in breast cancer [58,59]. Proliferation inhibition and apoptosis properties of ALA and AA have been suggested to be the result of different mechanisms of action (Figure 2).

There are many hypotheses for the mechanisms of the anti-proliferative effect of AA on cancer cells. However, the most probable among these is the generation of a significant amount of hydrogen peroxide, due to autoxidation of AA at such a high concentration (mM range) that preferentially targets cancer cells. Activation of the 2-oxoglutarate-dependent dioxygenase enzymes (2-OGDDs) is another proposed mechanism. The 2-OGDDs include the hydroxylases that regulate many processes including DNA demethylases. This DNA demethylation is particularly important, with recent investigations suggesting that AA has a role in regulating the ten-eleven translocase (TET) DNA demethylases in haematological cancers [42]. 

ALA’s anti-proliferative effect on tumour cells has been reported to be due to the increased reduction in epidermal growth factor receptor expression ErbB2 and ErbB3 [36,60]. Activation and overexpression of the previous receptors or their downstream effects have been described to be responsible for cell erratic growth and progression of malignancy in many cancer types, including breast cancer [61,62]. Previous studies showed that one of the mechanisms of ALA involved in cell death pathways is its ability to affect different downstream targets of the photolytic protein caspases. Caspases are photolytic proteins that upon activation induce intrinsic and programmed cell death [63,64]. ALA has been shown to be involved in the elevated activity of caspase-3 in breast cancer and neuroblastoma cell lines with a significant tumour size reduction in an animal model [65]. Prior to the increase in caspase-3 activity, it was also suggested that ALA plays a vital role in increasing p27, p21 and p53 protein expressions, which mediate cell cycle arrest and apoptosis through triggering caspase-3 activity as previously mentioned [66,67]. ALA not only increased the caspase protein family but it also down regulated the anti-apoptotic protein Mcl-1, Bcl-x_L_ and the epithelial–mesenchymal transition proteins in breast cancer cells [46,68,69].

Another proposed mechanism of apoptosis and cell death induced by ALA is its remarkable antioxidant effect, accompanied with the ROS scavenging effect, observed in breast cancer cell lines [17,65]. As previously stated, increased production of ROS has been recognised in cancer cell progression due to an increased rate of cell division, metabolic activity and malfunction of intrinsic antioxidant [17,48,65,70]. It was also reported that ALA induced autophagy via inhibition of O6-methylguanine-DNA methyltransferase (MGMT) protein, which consequently led to apoptosis and cytotoxicity to chemo-resistant colorectal cancer cells [71,72].

Therefore, it can be reported that as ROS scavengers, AA and ALA play two opposing roles: either as antioxidants or as pro-oxidants depending on the concentration and redox modulation. At small doses, they protect normal cells by eliminating reactive oxygen species. However, at a specific elevated threshold they can induce apoptosis and cytotoxicity to cancer cells [17,73,74,75].

## 6. Liposomes as a Drug Delivery System

Novel drug delivery systems, such as nanocarriers, are gaining remarkable attention due to their ability to improve the anticancer efficiency of various small molecules. During the past few years, solid lipid nanoparticles and liposomes have attracted much attention in the field of drug delivery [76]. They present some excellent material properties such as small particle size, biocompatibility, higher cellular uptake, improved efficacy, reduced dose and, in particular, easy functionalisation ability. The lipid coat of these systems can protect the encapsulated drugs from chemical degradation, improve their systemic circulation time and enhance their physical stability. In addition, these nanocarriers have been reported to modulate the release kinetics of the entrapped drug by delivering the drug in a controlled manner with less adverse effects and increased therapeutic efficacy [77,78].

Liposomes are the most commonly investigated nanostructure systems used in advanced drug delivery, which were first discovered by Bangham in 1963 [79]. Liposomes are established and promising systems for drug delivery and are believed to improve the effectiveness of the entrapped drugs; for example, they already used in clinics and the market to deliver cytotoxic drugs such as doxorubicin. Liposomes are lipid vesicles in the colloidal dimension in which aqueous medium is entrapped in phospholipid bilayer structures (Figure 3). The vesicles may consist of one (uni-lamella) or more (multi-lamella) lipid bilayers, with discrete aqueous spaces, and an internal aqueous core (Figure 3). They are well established for a range of pharmaceutical and biomedical applications. Due to the amphiphilic nature of phospholipids in aqueous media, liposomes are able to entrap both hydrophilic and lipophilic compounds [80], proteins and other macromolecules [81].

Tumours are often highly vascularised, but they have poor fluid circulation dynamics. Tumour vasculature is poorly formed and leaky. Poor perfusion combined with elevated interstitial fluid pressures within tumours decreases drug delivery and increases drug removal [82]. Conversely, the vascular properties of tumours enhance nanoparticle drug delivery through a process often referred to as the enhanced permeability and retention effect (EPR), whereby nanoparticles, with sizes less than the gaps in the tumour vasculature, escape the vasculature to accumulate in perivascular regions [72,83]. In other words, passive targeting of liposomes happens by transferring them into the tumour interstitia via leaky tumour vasculature through molecular movement within fluids. Thus, liposomes are providing an efficient means for delivering chemotherapeutic agents [77,83,84]. 

## 7. Liposomal Delivery of Ascorbic Acid

As previously stated, to achieve the required concentration at the target tissue, AA should be administered in high doses that are not usually tolerated by patients. Therefore, encapsulation of AA in liposomes with the aim of targeting cancer tissue, via enhanced permeability and retention properties, was another attractive alternative to reduce the dose. However, AA, with its high water solubility, was slightly loaded into the lipid vesicles with low entrapment efficiency. To overcome these problems, AA was chemically modified by esterification of the hydroxyl group with palmitic acid to obtain palmitoyl ascorbate (PA) (Figure 1c). PA is more lipophilic than AA; therefore, it can be easily partitioned into the bilayer membrane of the lipid vesicles with higher entrapment efficiency. In this way, it can cross the cell membrane more readily while maintaining the antioxidant efficiency of the parent AA. This allowed the use of smaller doses of the antioxidant that can be tolerated by patients [85,86,87]. Table 1 summarises the effects of AA within liposomal forms on cancer tissues based on literature.

A study was conducted comparing liposomal PA and a free injectable AA in a breast cancer model. The results indicated that liposomal PA at a much lower dose (liposomes equivalent to 20 mg/kg of PA injected intravenously every other day) was more effective than free AA (1 g/kg daily by intraperitoneal injection) in reducing the tumour growth rate of the 4T1 breast cancer model [86,88]. Other researchers combined PA with conventional chemotherapeutic agents to augment their therapeutic benefits. Paclitaxel liposomes were prepared with and without PA. Liposomes containing both paclitaxel and PA showed substantial antitumor effects in vivo and enhanced the activity of liposomal paclitaxel compared to PA-free liposomes. The co-treatment effect was clearly observed at a reduced dose of PA (10 mg/kg). Interestingly, PA-only liposomes showed significant antitumor activity. Therefore, liposomes provided a platform for enhancement of the antitumor activity of ascorbate [86].

A new method of active loading of epirubicin (EPI), a second-generation anthracycline antibiotic, using AA or an ammonium ascorbate gradient was investigated. This design gives the opportunity to load both the drug, EPI and AA or ammonium ascorbate into the vesicular structure (Figure 4). The method ensures rapid (less than 5 min) and efficient (about 100%) EPI encapsulation within the pegylated vesicles prepared using hydrogenated soy phosphatidylcholine, cholesterol and diglycerolphosphoethanolamine-polyethylene glycol (PEG) 2000 at a 5.5:4:0.5 ratio, respectively. The EPI encapsulation utilising the AA gradient resulted in a stable formulation both in vitro (no leakage for 360 days at 4°C) and in vivo (around 40% of the injected dose remained in the circulation after 24 h). The in vivo study showed high antitumor activity of liposomal EPI encapsulated with AA gradient against 4T-1 murine mammary cancer, a mouse model that is closely similar to human breast cancer. This novel liposomal dispersion improved EPI antitumour activity compared to that of the free EPI and liposomal containing EPI only. It was observed that the enhanced antitumor activity was a result of the synergistic antineoplastic activity of EPI together with AA. Another contributing factor to detect efficiency was the good solubility of EPI in the AA gradient that led to a better dissolution rate of the drug from liposomes and higher bioavailability [84].

Another study co-encapsulated docetaxel (DOC) and PA (palmitoyl ascorbate) in liposomes prepared using soybean phosphatidylcholine and cholesterol. The in vitro studies showed a synergistic antitumor effect in different cancer cell lines compared to either PA or DOC entrapped individually in liposomes. In vitro cytotoxicity studies demonstrated that the combined use of PA and DOC at a weight ratio of 200:1, respectively, had the highest synergistic effect when using HepG2, MCF-7 and PC-3 cell lines. The in vivo study confirmed the superiority of dual PA–DOC relative to PA or DOC individually [89]. The results indicated that PA, at a higher dose ratio, showed a substantial and greater effect than that of DOC liposomes without PA. Interestingly, PA liposomes were shown to be quite toxic to the cancer cell lines on their own and did not show a significant difference with DOC/PA200-liposomes. These findings are in good agreement with the previous study that reported the cytotoxic activity of PA liposomes [86,88]. Therefore, it was recommended that the co-delivery of PA and DOC in one nanocarrier system could provide a promising combined strategy for enhanced antitumour therapy [89].

Recently, a breakthrough in cancer therapy was the shift from non-specific delivery to strictly targeted therapy. This would increase the efficiency and reduce the side effects of the chemotherapeutic agents. Filipczak and co-workers [90] developed transferrin-targeted liposomes through targeting the vesicles to cells overexpressing transferrin receptors. The developed liposomes were triple loaded with a cocktail of ascorbic acid, mitoxantrone (MTX) and anacardic acid. Anacardic acid is a natural compound known for its potential anticancer activity [93]. Importantly, it inhibits the function of P-glycoprotein that is usually overexpressed on the surface of cancer cells and is responsible for the acquired resistance to current cytotoxic agents [94]. Accordingly, inclusion of anacardic acid would increase the sensitisation of the cancer cell towards the entrapped drug. The liposomal membrane consisted of hydrogenated phosphatidylcholine, dioleylphosphatidyl-ethanolamine, cholesterol, anacardic acid, 1,2-distearoyl glycerol-phosphatidylethanolamine-N-amino-poly-ethylene glycol 2000 and para-nitrophenol derivative DOPE-PEG (1,2-dioleoyl-sn-glycero-3-phosphoethanolamine-N-(polyethylene glycol)). The latter ingredient was added as a targeting moiety to the transferrin receptor. The liposomal formulations containing 5 mol% of anacardic acid with MTX and encapsulated by means of an ammonium ascorbate ion gradient exhibited a high degree of selectivity and toxicity towards melanoma cancer lines, but not to normal cells. The proposed mechanism was the possible generation of specific free radicals via an iron ions mechanism. Interestingly, the protective effect of ascorbate was observed [90].

## 8. Liposomal Delivery of Alpha Lipoic Acid

Many investigations were conducted combining the antioxidant effect of ALA with cytotoxic agents in solid lipid nanoparticles [47,72,95]. For liposomal formulations, improved therapeutic outcomes were reported when ALA was incorporated with ceftriaxone [96]. However, literature on liposomal delivery of ALA either alone or in combination with cytotoxic agents for cancer treatment is rare. The available information about the incorporation of ALA in liposomes mainly concerns improving the functionality of the developed nano-carrier system. 

It was reported recently that the use of docetaxel as a stand-alone therapy for breast cancer was accompanied by significant challenges such as suboptimal therapeutic response, chemoresistance (leading to more aggressive tumours), relapse and side effects due to a high dose requirement and drug toxicity [97]. Therefore, attention has been focused on developing a combination therapy using antioxidants, such as ALA, for improving the therapeutic outcome via synergism, overcoming drug resistance and reducing drug toxicity [98].

Stimuli-responsive liposomes have been recently developed with great success for their selective drug release at the target tissues, thus confining the unwanted side effects. The stimuli factor that could trigger the release of the loaded drug in liposomes could be either pH, temperature or ROS. Glutathione is an intracellular thiol-containing compound involved in regulating disulphide cleavable reactions [99]. As stated earlier, glutathione is overexpressed in tumour tissues. Because of the significant difference between the extra- and intra-cellular redox-environments and due to over-expression of glutathione in tumour cells, disulphide bonds (-S-S-) were studied as a responsive linker to design thiol-responsive liposomes. A cleavable disulphide bond responds to the reducing environment of cancer tissues that trigger the release of entrapped drug upon cellular uptake by cancer cells [100,101,102]. Table 1 shows the effects of ALA within liposomal forms on cancer cells based on available literature. 

As lipoic acid is a disulphide-containing compound (Figure 1b), it was explored in formulating a redox-responsive, thiol-containing vesicular drug delivery system. To maximise the anticancer efficiency of doxorubicin (DOX), the two antioxidants α-tocopheryl and lipoic acid were conjugated to produce two novel α-tocopheryl-lipoic acid conjugates (TL1 and TL2). Both conjugates were able to form stable vesicles on the nanoscale. Drug encapsulation efficiencies were found to be ∼60% and ∼55% with cumulative releases of ∼50% and ∼40% DOX, in response to the biological reducing agent glutathione, for TL1 and TL2, respectively (Figure 5). Both vesicles delivered DOX across HeLa cells in an efficient and significant manner more than that from the drug alone treatments [91].

Another novel conjugate of lipoic acid with phospholipids was recently synthesised. Dimeric lipoic acid-glycerophosphorylcholine (di-LA-PC) conjugate was produced and employed to prepare liposomes. To increase the serum stability, the obtained nanovesicles were crosslinked by the addition of a catalytic amount of the crosslinker dithiothreitol [92]. These liposomes were able to stabilise encapsulated DOX and showed a higher thiol-sensitive release of their payload upon cellular uptake by cancer cells. In vivo studies using a human breast carcinoma xenograft mouse model indicated their superiority compared to that of uncross-linked liposomes and DOX-loaded liposomes (which were similar in composition to the FDA approved Doxil^®^) [92].

## 9. Conclusions

This review highlighted the potential use of liposomes for the effective delivery of ascorbic acid and lipoic acid, as examples of strong antioxidants. Hence, liposomes are promising nanocarriers to deliver antioxidants in considerable effective concentrations to the cancer tissue. Some studies demonstrated that liposomes with antioxidants can stand alone as effective anticancer agents. This is in addition to the confirmed synergistic effects of antioxidants with traditional anticancer agents. Furthermore, some antioxidants can impart functionality to the liposomes that can further augment cell death by the entrapped liposomes. It is time to incorporate those antioxidants into liposomes to be manufactured and used in clinics for the patients’ benefits.

## Figures and Tables

**Figure 1 antioxidants-09-00359-f001:**
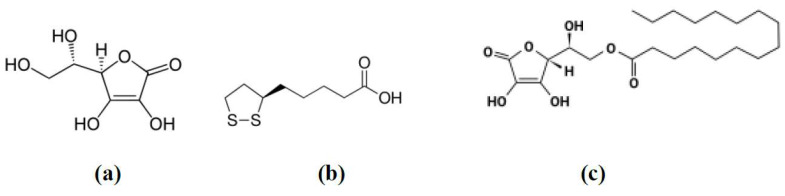
Chemical structures of ascorbic acid (**a**), lipoic acid (**b**) and ascorbyl palmitate (**c**).

**Figure 2 antioxidants-09-00359-f002:**
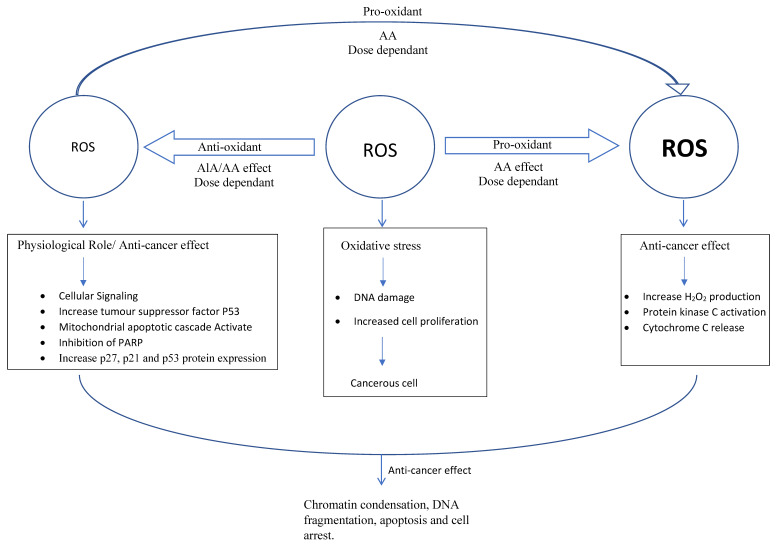
Schematic illustration for antioxidant (AA: ascorbic acid; ALA: alpha lipoic acid) effects on reactive oxygen species (ROS) and/or mechanism of action in cancer cells.

**Figure 3 antioxidants-09-00359-f003:**
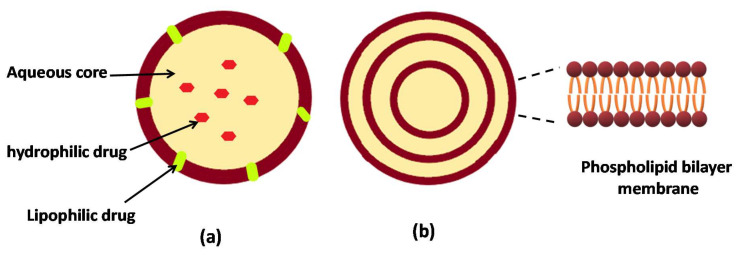
Schematic representation of unilamellar (**a**) and multilamellar (**b**) liposomes.

**Figure 4 antioxidants-09-00359-f004:**
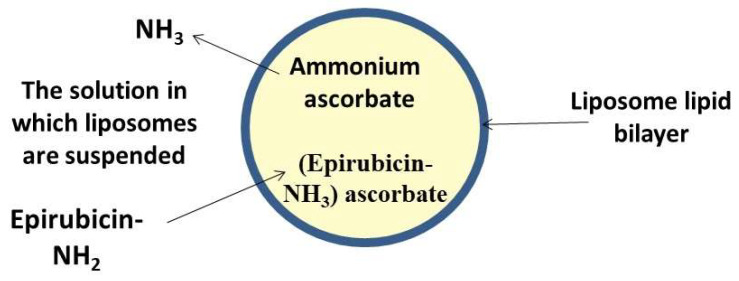
Schematic representation of epirubicin encapsulation within liposomes via indirect encapsulation—ammonium ascorbate gradient.

**Figure 5 antioxidants-09-00359-f005:**
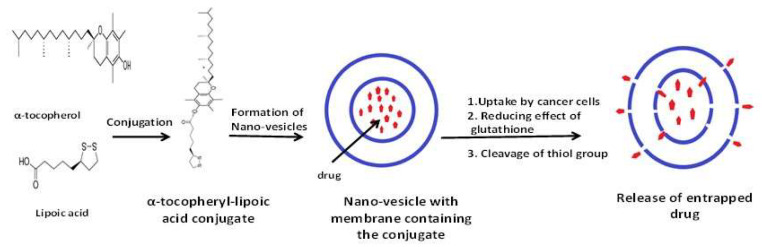
Schematic illustration of the proposed mechanism of action of thiol-triggered nano-vesicles using α-tocopherol-lipoic acid conjugate.

**Table 1 antioxidants-09-00359-t001:** The effects of antioxidants, ascorbic acid and lipoic acid in liposomal forms on cancer cells.

Antioxidant	Active Ingredient	Effect/Cancer Model	Formulation	Outcomes	Reference
Ascorbic acid (AA)	Epirubicin	Anticancer/murine 4T-1 breast cancer model	Pegylated epirubicin liposomes via AA gradient versus free epirubicin and epirubicin only liposomes.	Liposomal epirubicin encapsulated with AA gradient showed higher antitumor activity than liposomal with only epirubicin and free drug.	[84]
Palmitoyl ascorbate (PA) and ascorbic acid (AA)	−	Anticancer/Breast cancer model	Palmitoyl ascorbate liposomes versus free ascorbic acid.	Liposomal PA was more effective than free AA.	[88]
Palmitoyl ascorbate (PA)	Docetaxel	Anticancer/liver, (HepG2), breast (MCF-7) and prostate (PC-3) cancer cell lines	Combined encapsulated liposomes with PA and docetaxel versus PA or docetaxel entrapped individually in liposomes.	Co-delivery of PA and the drug in the liposomal system enhanced the antitumour therapy.	[89]
Ascorbic acid	Mitoxantrone	Anticancer/melanoma cancer lines	Liposomes with triple ingredients: ascorbic acid, mitoxantrone and anacardic acid.	High degree of specificity and toxicity towards cancer lines and not normal cells.	[90]
α-tocopheryl—lipoic acid as two novel conjugates	Doxorubicin	Anticancer/HeLa cells	α-tocopheryl and lipoic acid conjugated liposomal nano-vesicular systems encapsulated drug versus drug alone.	Both types of vesicles delivered the drug across HeLa cells in more effective way than the drug alone.	[91]
Lipoic acid as dimeric lipoic acid-glycerophosphorylcholine conjugate	Doxorubicin	Anticancer/breast carcinoma	Dimeric lipoic acid-glycerophosphorylcholine conjugated nano-vesicular liposomes (cross-linked with dithiothreitol) encapsulated doxorubicin versus uncross-linked liposomes and drug loaded liposomes (with no antioxidant conjugate)	Cross-linked liposomes showed superior cellular uptake by cancer cells than uncross-linked liposomes and drug loaded liposomes.	[92]

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
