# Peer review of "An Overview of the Antioxidant Effects of Ascorbic Acid and Alpha Lipoic Acid (in Liposomal Forms) as Adjuvant in Cancer Treatment"

_antioxidants, 2020, doi:10.3390/antiox9050359_

Round 1
Reviewer 1 Report
In this manuscript, the anticancer effects of AA and ALA in liposomal forms are well-summarized. If the authors tabulate several parts of the manuscripts, it will enhance understanding of readers. Please present tables which compare the anticancer effects among AA or ALA alone, AA or ALA delivered by liposomes, and AA or ALA with liposome and chemotherapeutic agent to effectively summarize the lines 141-323 and help understanding of readers.
Author Response
Thank you for the valuable comments. These comments helped us to improve the quality of our manuscript. The manuscript has been revised taking into consideration these comments and the changes made to the manuscript are presented as green highlights. The proceeding sections present the response to each comment and the changes made in response to each comment.
Comment:
In this manuscript, the anticancer effects of AA and ALA in liposomal forms are well-summarized. If the authors tabulate several parts of the manuscripts, it will enhance understanding of readers. Please present tables which compare the anticancer effects among AA or ALA alone, AA or ALA delivered by liposomes, and AA or ALA with liposome and chemotherapeutic agent to effectively summarize the lines 141-323 and help understanding of readers.
Response:
Table 1 was added, please refer to Table 1.
Reviewer 2 Report
The review manuscript by Attia et al. summarized the use of antioxidants in liposomal form for cancer therapy. The authors well captured recent important discoveries in the field that are relevant to the topic. Overall, this manuscript was well organized, and following are specific suggestions to improve the quality of the manuscript.
- What may improve this review is citing the original work rather than other review articles.
- There are many sentences lacking appropriate reference. For example, from line 43 to line 45, the authors should cite the original work addressing that oxidative stress could result in diabetes, Parkinson's disease, Alzheimer's diseases, acute renal and lung failure, and cancer one by one.
- In line 229, the authors used term “PA”. What does “PA” mean?
- As the authors mentioned, the action mechanism of ascorbic acid (AA) seems to be dependent on the dose of administration. In addition to the effect of low concentration of ascorbic acid, it would be better to state the effect of high concentration of ascorbic acid.
- In line 122, the authors mentioned that dehydroascorbate, the oxidized form of AA, is transported into cells via the glutathione-1 glucose transporter. Please provide the reference of this sentence.
- In line 157, the authors stated that “Proliferation inhibition and apoptosis properties of ALA and AA have been suggested to be the result of different mechanism of actions”. Although the action mechanism of ALA was described in detail, the precise mechanism of AA was not explained well.
- In line 181, the sentence “Where it can induce the metastasis process and acts as an antiapoptotic agent” is incomprehensible.
- The paragraph from line 292 to line 301 seems not to be matched to the section title.
- Inclusion of additional figures would help readers to understand the manuscript better. For example, illustration of the action mechanisms of AA and ALA in cancer cells, the scheme of encapsulation method utilizing AA gradient, and so on.
- This manuscript would benefit from English editing.
Author Response
Thank you for the valuable comments. These comments helped us to improve the quality of our manuscript. The manuscript has been revised taking into consideration these comments and the changes made to the manuscript are presented as green highlights. The proceeding sections present the response to each comment and the changes made in response to each comment.
Point 1: What may improve this review is citing the original work rather than other review.
Response 1: Done please see the additional references. Reference adjusted accordingly
Point 2: There are many sentences lacking appropriate reference. For example, from line 43 to line 45, the authors should cite the original work addressing that oxidative stress could result in diabetes, Parkinson's disease, Alzheimer's diseases, acute renal and lung failure, and cancer one by one.
Response 2: Done, please refer to new added references 2-7.
Point 3: In line 229, the authors used term “PA”. What does “PA” mean?
Response 3: PA is the abbreviation to Palmitoyl Ascorbate (added).
Point 4: As the authors mentioned, the action mechanism of ascorbic acid (AA) seems to be dependent on the dose of administration. In addition to the effect of low concentration of ascorbic acid, it would be better to state the effect of high concentration of ascorbic acid.
Response 4: The effect of higher concentration of ascorbic acid was clarified. Please refer to lines 125-130.
Point 5: In line 122, the authors mentioned that dehydroascorbate, the oxidized form of AA, is transported into cells via the glutathione-1 glucose transporter. Please provide the reference of this sentence.
Response 5: The reference was added (reference 42).
Point 6: In line 157, the authors stated that “Proliferation inhibition and apoptosis properties of ALA and AA have been suggested to be the result of different mechanism of actions”. Although the action mechanism of ALA was described in detail, the precise mechanism of AA was not explained well.
Response 6: The mechanism of action of the anti-proliferation effect of ascorbic acid to cancer cells was added. An illustrated diagram was also included (Figure 2), Figures’ numbers were adjusted accordingly.
Point 7: In line 181, the sentence “Where it can induce the metastasis process and acts as an antiapoptotic agent” is incomprehensible.
Response 7: the sentence was corrected and the whole paragraph was rephrased. Please refer to lines 191-194.
Point 8: The paragraph from line 292 to line 301 seems not to be matched to the section title.
Response 8: This section is an introductory to justify the use of lipoic acid, as a thiol containing compound, to formulate thiol-triggered release of intra-liposomal dugs. This section was rearranged and Figure 5 was constructed for clarity and to make it easier for the reader to follow. The references of this section was rearranged accordingly.
Point 9: Inclusion of additional figures would help readers to understand the manuscript better. For example, illustration of the action mechanisms of AA and ALA in cancer cells, the scheme of encapsulation method utilizing AA gradient, and so on.
Response 9: Graphical abstract and figures 2, 4 and 5 were added.
The manuscript was reviewed and the English was corrected throughout the text.
Round 2
Reviewer 2 Report
Most of my concerns seem to be clearly resolved. Especially, addition of Table 1, Figure 2, and Figure 5 would help readers to understand the manuscript better. Following are two minor comments.
- In Figure 2, the authors presented three circles, and all circles look same. What about changing these circles to different expressions such as “low ROS”, “intermediate ROS”, “high ROS”?
- I could not find the evidence that the oxidized form of AA is transported into cells via the glutathione-1 glucose transporter in reference 42 which the authors mentioned. I even could not find the gene named “glutathione-1 glucose transporter”. What is official name of this gene? Please use official term in the manuscript.
Author Response
Many thaks for the constructive minor comments, here is the response:
For point 1: Low ROS, Medium ROS, High ROS have been identified on Figure 2.
For point 2: The official term of the gene has been mentioned and one reference (41) has been replaced to clariy the sentences. Please refer to the red font within lines 125 and 126.